# Investigating Immunomodulatory Biomaterials for Preventing the Foreign Body Response

**DOI:** 10.3390/bioengineering10121411

**Published:** 2023-12-11

**Authors:** Alexia Kim, Mauricio A. Downer, Charlotte E. Berry, Caleb Valencia, Alex Z. Fazilat, Michelle Griffin

**Affiliations:** Department of Surgery, Division of Plastic and Reconstructive Surgery, Stanford University School of Medicine, Stanford, CA 94305, USA; alexiaki@usc.edu (A.K.); mdown18@stanford.edu (M.A.D.); berryc@stanford.edu (C.E.B.); afazilat@usc.edu (A.Z.F.)

**Keywords:** biomaterials, foreign body response, macrophages, immunomodulation

## Abstract

Implantable biomaterials represent the forefront of regenerative medicine, providing platforms and vessels for delivering a creative range of therapeutic benefits in diverse disease contexts. However, the chronic damage resulting from implant rejection tends to outweigh the intended healing benefits, presenting a considerable challenge when implementing treatment-based biomaterials. In response to implant rejection, proinflammatory macrophages and activated fibroblasts contribute to a synergistically destructive process of uncontrolled inflammation and excessive fibrosis. Understanding the complex biomaterial–host cell interactions that occur within the tissue microenvironment is crucial for the development of therapeutic biomaterials that promote tissue integration and minimize the foreign body response. Recent modifications of specific material properties enhance the immunomodulatory capabilities of the biomaterial and actively aid in taming the immune response by tuning interactions with the surrounding microenvironment either directly or indirectly. By incorporating modifications that amplify anti-inflammatory and pro-regenerative mechanisms, biomaterials can be optimized to maximize their healing benefits in harmony with the host immune system.

## 1. Introduction

Biomaterials have extensive applications in the field of medicine, serving as vital components in various medical interventions and treatments. At the forefront of regenerative medicine, these applications include implants, drug delivery systems, wound dressings, diagnostic tools, surgical devices, and bioabsorbable materials [1,2]. However, it is important to acknowledge that the use of biomaterials can result in implant rejection and trigger an inflammatory and profibrotic reaction known as the foreign body response (FBR), which compromises the functionality and long-term performance of the biomaterial used and can lead to failure of tissue regeneration [1].

Tissue regeneration can be compromised due to various factors, including inflammation, extracellular matrix dysfunction, angiogenesis deficiency, mechanical stress, and fibrosis [3]. Inflammation is a pivotal player in tissue repair and regeneration; however, an exaggerated or prolonged inflammatory response can result in scarring and hinder the regenerative process [3]. The extracellular matrix (ECM) provides critical structural support to tissues and is indispensable for successful regeneration. A dysfunctional ECM can disrupt cell migration, proliferation, and differentiation, ultimately impeding wound healing. Adequate angiogenesis, the formation of blood vessels, is vital for supplying tissues with the necessary nutrients and oxygen during regeneration. Insufficient angiogenesis can hamper tissue health and lead to reduced healing potential [3]. Fibrosis, characterized by the excessive formation of fibrous tissue in response to injury, can obstruct normal tissue regeneration by forming excessive scar tissue [3]. FBR contains various aspects of tissue regeneration failure, encompassing the factors mentioned above [1].

Therefore, to combat the FBR, a better understanding of the fundamental mechanisms of fibrotic capsular formation around an implant is crucial for developing effective treatments. Macrophages play an important role in material recognition and are often found adhering to the surrounding surfaces of the foreign object [4]. Macrophages communicate with other cell types in the wound environment through the secretion of cytokines. Cytokines are necessary for the signaling pathways involved in the FBR. Among these signaling molecules, transforming growth factor β (TGF-β) is a particularly involved cytokine secreted by macrophages [4]. Extensive research has shown that TGFβ facilitates tissue fibrosis through various mechanisms [5].

To minimize or prevent FBR, biomaterial design has recently evolved with the aim of either decreasing or altering the immune response and ultimately evading activation of the fibrotic cascade [6]. Key aspects of biomaterial properties, such as surface chemistry, topography, and material composition, have received significant attention in relation to FBR. For instance, researchers have discovered that the surface texture of breast implants affected the thickness of the fibrotic encapsulation that surrounded the implants [7]. Another strategy involves utilizing an acellular dermal matrix, which is a matrix devoid of cells and consists only of an extracellular matrix [8]. The dermal matrix provides a structural scaffold that promotes angiogenesis and regeneration without containing any cellular antigens. Another area of growing interest involves protein-functionalized and growth factor-releasing biomaterials, which allow for the material to actively interact with the host immune system and surrounding microenvironment. The utilization of hydrogels that suppress specific cytokines has also emerged as a widely adopted and successful approach for reducing an FBR [9].

To summarize, the advancements in biomaterial design have greatly contributed to the reduction of inflammation and FBR. It is essential to consider these strategies when designing increasingly intricate implant devices. This review aims to provide a comprehensive explanation of the techniques used to overcome the immune response and modify the FBR for improved outcomes.

## 2. Foreign Body Response (FBR)

The FBR is a complex series of biological events that occur when a biomaterial or foreign object is implanted into the body. While the exact sequence and duration of these events may vary depending on factors such as the type of biomaterial and the individual response, the FBR represents an overall inflammatory process involving protein adsorption, neutrophil and macrophage activation, foreign body giant cell formation, and fibrotic encapsulation surrounding the implanted biomaterial (Figure 1) [10,11]. While these stages are dynamic and may occur concurrently, each has been recorded consistently in the formation of FBR [10,11,12]. When a biomaterial is surgically introduced into the body, an immunological cascade is activated. The biomaterial may be a medical device, implant, scaffold, or any foreign object designed for biomedical applications [10,11]. Upon contact with the surrounding tissue and blood, the biomaterial interacts with the host plasma components, leading to the adsorption of proteins, such as fibrinogen, albumin, and immunoglobulins, onto its surface [11].

Ibrahim et al. provided evidence that the FBR exhibits a degree of specificity depending on the type of implant employed, resulting in variable levels of capsule thickness and cellular reactions [12]. In their research, they inserted biocompatible materials including silicone sheets, polyvinyl alcohol (PVA), expanded PTFE (EPTFE), Polypropylene and cotton sheets (control materials) in order to investigate whether the use of biocompatible materials could help reduce the occurrence of FBR [12]. This experiment demonstrated the capacity to amplify or diminish FBR-associated cellular activity. Notably, their specific findings indicated an increase in both cellular activity and capsular thickness when using cotton sheets. Silicone sheets formed the thinnest capsules, and PVA showed an increased number of Giant cells. Histological analysis using Masson’s trichrome stained sections demonstrated a non-statistically significant change between the implants. Consequently, they concluded that meticulous consideration of biomaterial selection is essential to mitigate the foreign body response in biomaterial implantation. This example bears relevance to the realm of plastic surgery procedures involving silicone implantation [12].

The physicochemical characteristics of the biomaterial, including surface chemistry and charge, influence this adsorption process [10,11]. Subsequently, complement proteins are activated in reaction to the foreign object, mediating the recruitment and adhesion of additional immune cells. As neutrophils and macrophages adhere to the surface of the biomaterial, an acute inflammatory response is initiated.

### 2.1. Acute Inflammatory Response

Following the implantation of a biomaterial, neutrophils promptly adhere to the surface, initiating the inflammatory cascade. These innate immune cells demonstrate phagocytic capabilities, engulfing and eliminating foreign particles, debris, and potential pathogens associated with the biomaterial [13]. Additionally, neutrophils release an array of antimicrobial molecules, including reactive oxygen species (ROS) and antimicrobial peptides, which enable neutrophils to effectively eradicate potential threats and ensure tissue sterility [11]. In addition to their contribution to inflammation, neutrophils can secrete a variety of growth factors and matrix metalloproteinases (MMPs) to promote ECM remodeling [14]. Within 48 h of neutrophil activity, neutrophils are gradually replaced by macrophages. With the onset of macrophages, the initial inflammatory response transitions to an FBR [13].

### 2.2. Role of Macrophages in FBR

Macrophages are key contributors to FBR and play a crucial role in driving the overall immune response to implanted biomaterials. They possess autonomous capabilities by continuously proliferating and releasing chemotactic factors that recruit additional macrophages [13]. Moreover, these immune cells exert control over the immune response by undergoing polarization, which is the process by which macrophages adopt distinct functional phenotypes based on their local environment [15]. Classically activated (M1) macrophages exhibit proinflammatory properties and play a primary role in orchestrating the FBR [15] by releasing tumor necrosis factor-alpha (TNF-α), interleukin (IL)-1β, IL-6, and IL-8 (Table 1) [11,14,16,17,18,19,20,21,22]. These proinflammatory mediators are vital for wound healing and drive macrophage-mediated inflammation, collectively amplifying the FBR [14]. In contrast, alternatively activated (M2) macrophages coordinate an anti-inflammatory response by releasing IL-13, IL-10, and IL-4 and are prominently involved in tissue remodeling [15].

Phagocytosis is a fundamental function of macrophages, enabling them to engulf and eliminate debris, foreign particles, and potentially viable cells in the vicinity of biomaterials. This process is triggered by the recognition of specific molecular patters by pattern recognition receptors (PRRs) present on the surface of macrophages [14]. PRRs, including toll-like receptors, NOD-like receptors, and scavenger receptors, enable the identification of pathogen-associated molecular patterns (PAMPs) derived from microorganisms, such as bacteria and fungi, and damage associated molecular patterns (DAMPs) that are released during tissue injury, inflammation, and cell death [14]. By interacting with the PRRs expressed by macrophages, PAMPs and DAMPs in the surrounding wound environment induce the phagocytic activity of macrophages [14].

However, insufficient degradation of a biomaterial by phagocytosis may lead to the formation of foreign body giant cells (FBGCs) [23]. FBGCs are large multinucleated macrophages that form from the fusion of multiple macrophages in response to foreign objects or implanted biomaterials [23]. FBGCs can adhere to the surface of biomaterials and release enzymes such as acid hydrolases, proteases, and reactive oxygen species, which can lead to chronic inflammation and impair the functionality of the biomaterial [23].

Macrophages may also further amplify the inflammatory response by activating transcription factors like nuclear factor Kappa B (NFK- β), which stimulates the production and release of additional proinflammatory cytokines and chemokines [24]. Recruited by these signaling molecules, fibroblasts are activated at the implantation site, inducing the formation of a fibrous encapsulation [25].

### 2.3. Fibrous Capsular Formation in FBR

During the process of normal wound healing, various cells within the wound environment, such as macrophages, platelets, and adipocytes, secrete growth factors that promote the recruitment and activation of fibroblasts [25]. These growth factors, including platelet-derived growth factors (PDGF), TGF-β, and vascular endothelial growth factors (VEGF), play a crucial role in stimulating the replacement of weak fibrin cells with a stronger collagenous extracellular matrix (ECM) [25].

During the FBR, macrophages release profibrotic growth factors, causing fibroblasts to proliferate and migrate to the biomaterial–tissue interface [25]. At the implant site, fibroblasts differentiate into myofibroblasts and commence excessive synthesis and deposition of collagen fibers, gradually forming a dense network that surrounds the biomaterial [25]. The thick fibrous capsulation isolates the biomaterial from the surrounding tissue environment, which interferes with proper wound healing, prevents integration of the biomaterial within the tissue, and impairs biomaterial compatibility and efficacy [25].

Macrophage-mediated fibrous capsulation is a topic of ongoing investigation due to its significant impact on implant viability. In a study conducted by Kim et al. [15], the researchers explored the effects of macrophage polarization on reducing capsular thickness associated with silicone breast implants [15]. Silicone implants were coated with IL-4, a cytokine that promotes activation of M2 macrophages, prior to implantation in mice. The IL-4-coated implants led to an upregulation of mannose receptor arginase-1, which is primarily expressed by M2 macrophages, and an overall reduction in inflammation and fibrous capsulation [15]. Ongoing research aims to investigate the relationship between the immune system and biomaterial design to mitigate the FBR associated with biomaterials.

### 2.4. Biomaterials in Surgery

In the field of plastic and reconstructive surgery and skin regeneration, a wide array of biomaterials finds application in reconstructive procedures [26]. Peng et al. conducted a comprehensive review encompassing various biomaterials utilized in plastic and reconstructive surgery. Among these materials, natural biomaterials comprise components such as animal-derived proteins, decellularized tissues, collagen, and hyaluronic acid. While natural biomaterials tend to evoke a minimal FBR, their practical application may be limited in scope or accompanied by high costs [26].

Conversely, synthetic biomaterials are also employed extensively in surgical settings for reconstructive purposes [26]. These materials encompass, but are not restricted to, substances like silicone, expanded polytetrafluoroethylene, polymethyl methacrylate, polylactic acid, and polyglycolic acid. Synthetic materials offer a broad spectrum of applications, including prostheses, 3D printing, sutures, and drug delivery. However, they carry the potential for eliciting a foreign body response such as capsular formation, biomaterial degradation, infection, and granuloma formation [26].

Presently, researchers are actively engaged in efforts to integrate various materials alongside biomaterial implants in order to mitigate the immune response they may trigger [12].

## 3. Immune System and Biomaterial Design

Advancements in biomaterial design are able to overcome implant rejection and promote tissue integration through tunable properties that either render the biomaterial immunologically inert or enhance their immunomodulatory capabilities. Biomaterials can be synthesized in a wide range of compositions, sizes, shapes, and porosities with surface functionalization specifically catered to the appropriate disease context, allowing for optimization and customization of design [27,28]. By carefully designing implantable materials and surfaces with biocompatible parameters, biomaterials can be implemented in a way that maximizes their therapeutic potential and promotes wound healing in diverse contexts.

Specific modifications can significantly reduce FBR and evade immune recognition. Generally, biomaterials with decreased protein adsorption are developed to prevent cell adhesion and activation [29,30,31,32] (Table 2). Protein adsorption represents the crosstalk at the biomaterial–tissue interface that initiates the immune response and FBR. While monocytes and macrophages tend to adsorb on implant surfaces, the protein layer chemistry can be modified to target a specific amount and identity of the substrates that are adsorbed [30]. Surfaces with poly(ethylene glycol) (PEG), for instance, minimize interactions at the interface, resulting in a significantly thinner fibrotic capsule surrounding the implant [31].

Researchers have explored various methods to reduce biofilm formation in the context of biomaterial implants. Biofilms, structured communities of microorganisms that can attach to implant and medical device surfaces, act as protective layers for these microorganisms on the implanted material. This protection can lead to prolonged inflammation, infections, and complications, thereby intensifying the FBR. Zhi et al. showcased a reduction in biofilm formation on silicone rubber surfaces when coated with polyhexanide (PHMB) and higher molecular weight PEG, highlighting the anti-fouling properties of these coatings [33].

While inert biomaterials may be particularly useful for removable or temporary implants, integration within the tissue microenvironment is ideal for biomaterial implants that provide sustained benefits and promote wound healing [29]. By actively participating in pro-healing mechanisms and coordinating positive crosstalk with immune cells at the interface, tissue integration can both protect and amplify the therapeutic benefits of biomaterials while preventing implant rejection and fibrotic encapsulation [29]. Certain material properties that aid in tissue integration may enhance the anti-inflammatory host immune response by shifting macrophage polarity from an M0 or M1 phenotype to an M2 phenotype or by promoting anti-inflammatory cytokine production while suppressing pro-inflammatory cytokine production [29]. The integration of biomaterials also involves robust and accelerated wound repair and regeneration, resulting in increased cell infiltration and proliferation, vascularization, and functional tissue formation [29]. By preventing FBR through active participation in anti-inflammatory and pro-regenerative mechanisms, advancements in tunable properties of biomaterials, such as surface chemistry, topography, protein functionalization, growth factor supplementation, and cell therapy, may enhance the therapeutic benefits of implanted biomaterials (Figure 2).

## 4. Surface Chemistry

Surface chemistry plays a crucial role in determining the interactions of the biomaterial and the surrounding cells, tissues, and fluids. It influences various aspects, such as adsorption, cell adhesion, and biocompatibility [34]. The surface chemistry of biomaterials may be modified through chemical modifications, physical modifications, and radiation. Commonly, biomaterial surfaces are functionalized with a coating of grafting of bioactive molecules, such as functional groups or cytokines [34]. These moieties can alter surface charge, hydrophilicity, and hydrophobicity. Agarwal et al. [35] investigated the use of modified polymeric films in wound healing. By adding silver to the polymeric films, the antimicrobial characteristics of the biomaterial were enhanced, allowing for improved wound healing without risk of infection and cytotoxicity [35]. Soto et al. investigated the release of nitric oxide (NO) from biomaterials and its effect on the FBR in diabetic swine. Research showed that NO-releasing biomaterials can counteract the FBR response in diabetic swine while initiating tissue regeneration [36].

### 4.1. Topography

The topography of a biomaterial refers to the roughness of a surface (asperity and waviness), and changes in topography can shape the growth of surrounding tissue, cell adhesion, and cell morphology [37]. Biomaterial topography can vary across several parameters, including ordered/disordered, aligned/non-aligned, and patterned/unpatterned. The discovery that biomaterial surface topography and nanotopography (surface texture on a nanoscopic scale) affects cellular adhesion, proliferation, cell function, stem cell differentiation, morphology, and tissue integration has prompted the investigation of topography effects on the immune response, wound healing, and cellular uptake [38,39,40,41,42].

Understanding that macrophages act as key mediators of the FBR, modifications to the topography of biomaterials have yielded promising and intriguing results in influencing the M1/M2 polarization and functional behavior of macrophages [43].

Changes in surface texture have been shown to alter the secretion of pro-inflammatory cytokines such as interleukins (IL-1β, IL-6, TNF-α) and chemokines (MCP-1, MIP-1α) [44]. In particular, increasing surface roughness has produced an increase in macrophage inflammatory protein-1α (MIP-1α) in vitro [45]. In addition to inflammatory markers, topography modifications have also influenced the expression of regenerative markers where rougher titanium surfaces resulted in an increased expression of BMP-2 by macrophages, indicating the potential for increased bone formation [46].

Furthermore, surface texture has also been found to modulate macrophage adherence and spreading, though this behavior varies across biomaterial and tissue type. In some contexts, macrophages have been shown to preferentially attach to smooth substrates over those that are rougher. However, in a study using pure titanium with textural surface differences, adhesion and spreading of macrophages were shown to increase in vitro when rougher surfaces were applied [46].

In the context of wound healing, cytoskeleton formation and macrophage adhesion can be altered to promote the healing response by modifying biomaterial surface texture [47]. In a study by Bota et al., changes in roughness were shown to alter cell response in vitro and tissue integration in vivo [47]. Upon subcutaneous implantation in mice, increased texture on a porous polyvinyl alcohol implant resulted in improved wound healing through enhanced vascularity and reduced fibrous capsule formation compared to the non-textured implant [47].

### 4.2. Biomaterial Composition

Biomaterial composition has led to advancements in decreasing the FBR. Kyriakides et al. identified distinct subtypes of macrophages that exhibit different fibrotic and regenerative responses when exposed to synthetic- or natural-based polymer biomaterials, highlighting the direct influence of material properties on the interaction between immune cells and biomaterials [48]. While certain material compositions have been shown to elicit less drastic inflammation and FBR, acellular biomaterials, or more specifically, acellular dermal matrices (ADM), have also demonstrated diminished FBR and improved healing [49]. These biocompatible ECM scaffolds have recently been found to both decrease the inflammatory response and reduce multinucleated giant cell formation, suggesting that ADM may be a viable biomaterial composition for preventing FBR and promoting tissue integration [49].

### 4.3. Protein-Functionalized Biomaterials

The use of bioactive peptides can actively mitigate FBR through immunomodulatory mechanisms and aid in the integration of biomaterials by promoting angiogenesis and tissue regeneration [29,50,51]. Biomaterials functionalized with ECM-derived peptides allow for more precise and controlled mechanisms of promoting anti-inflammatory M2 macrophage polarization and influencing cytokine production [52,53,54]. Overall, peptide-functionalized materials not only have the capacity to modulate the immune response, but are also able to replicate a more natural healing microenvironment [29].

Integrin-targeting peptides directly interact with immune cells in the wound environment to drive macrophage polarity towards an anti-inflammatory phenotype and shape multiple upstream signaling pathways [52]. Cha et al. observed that inhibition of integrin α2β1 blocks the induction of anti-inflammatory M2 macrophages polarization. This suggest that integrin peptides play an integral role in promoting a microenvironment that favors M2 polarity, reduces fibrosis, and enhances biocompatibility [52].

Integrins including arginine-glycine-aspartic acid (RGD) peptides are well known to influence cell-cell interactions around biomaterials. [52,53,54] Specifically, integrin-binding arginine-glycine-aspartic acid (RGD) peptides, which are derived from fibronectin, have recently been integrated into biomaterial engineering as common immunomodulatory components [55]. Wu et al. utilized phosphatidylserine-containing liposomes (PSLs) with RGD motifs to drive M2 polarization of macrophages and promote bone regeneration [54]. Overall, RGD-PSLs had a concerted immunomodulatory effect through enhanced M2 macrophage marker genes such as Arg-1, FIZZ-1, and YM-1, and suppression of pro-inflammatory cytokine (IL-1β, IL-6, TNF-α) expression both in vitro and in vivo using a calvarial defect rat model [54].

While RGD-PSLs had significant immunomodulatory and regenerative effects, Wang et al. has also shown how RGDs alone can drive M2 macrophage polarization and promote anti-inflammatory mechanisms through integrin interactions [53]. RGD adhesive peptides were released via photodegradative alkoxylphenacyl-based polycarbonate (APP) nanocomposites, allowing for the researchers to observe the user-controlled integrin activation of macrophages [53]. Through the RGD-induced activation of the αvβ3 integrin expressed on the surface of macrophages, decreased expressions of proinflammatory cytokines TNF-α and IL-6 were observed along with increased expressions of anti-inflammatory cytokines TGF-β and IL-10, suggesting a transition from an M1 to an M2 macrophage phenotype [53]. By utilizing a user-controlled biomaterial in conjunction with peptide functionalization, the biomaterial–cell response can be optimally tuned to be more reparative and regenerative.

### 4.4. Growth Factor-Releasing Biomaterials

By binding to their corresponding receptors and activating specific signal transduction pathways, growth factors directly modulate inflammatory mechanisms, drive granulation tissue formation, and promote angiogenesis [56,57] (Table 3). In the context of wound healing, they are also critical for ECM formation and remodeling [56,57]. Unfortunately, growth factors have short half-lives and easily degrade in the wound microenvironment [56]. Different biomaterials address this challenge by providing a stable and biocompatible platform that both sustains and maximizes the therapeutic advantages of growth factors. Growth factor-releasing biomaterials are promising advancements that directly facilitate regenerative processes and prevent an overactive inflammatory response.

Using adipose-derived stem cells (ASCs) and endothelial progenitor cells (EPCs), Wu et al. highlighted platelet-derived growth factor AA (PDGF-AA) as an essential growth factor that is necessary for the angiogenic and regenerative effects of ASCs and EPCs [58]. When PDGF-AA was knocked down in ASC cell lines, wound healing was delayed in skin-wound areas with PDGF-AA knockdown ASCs [58]. In response to nanofibrous scaffolds with recombinant human epidermal growth factor (rhEGF), faster wound closure rates were observed compared to murine excisional wounds without rhEGF [56,59]. Several studies have shown that basic fibroblast growth factor (bFGF) promotes cell proliferation, accelerated wound contraction, neovascularization, and enhanced re-epithelialization with robust granulation tissue formation [60,61,62]. Soderlund et al. [63] demonstrated that synthetic glycosaminoglycans could release bFGF slowly, resulting in minimized FBR and a higher ratio of M2:M1 macrophages, altering the immunomodulatory capabilities of growth factor-releasing biomaterials.

### 4.5. Cell Therapy

The field of cell therapy is rapidly evolving and holds great promise in addressing the FBR associated with biomaterials. Recent research has explored the utilization of human mesenchymal stem cells (hMSCs) in constructing devices. In these studies, the addition of hMSCs demonstrated enhanced integration and vascularization [64]. The integration of hMSCs is of particular interest as these stem cells are found to secrete high levels of anti-inflammatory factors such as IL-1 receptor antagonist [65].

Recognizing the significance of the interaction between immune cells and the cells residing within the biomaterial is crucial when trying to minimize the immune response. The interaction of macrophages is a subject of special interest, as these cells are frequently observed adhering to foreign objects and play crucial roles in immune signaling [4]. A research experiment investigated the relationship between the cells within a biomaterial and the resulting FBR. The researchers conducted this experiment by seeding autologous cells onto a decellularized ECM known for its regenerative properties. Surprisingly, the matrix containing cells resulted in a fibrotic healing response [66].

## 5. Future Directions

As researchers delve deeper into the realm of biomaterials and their applications in medical contexts, it remains crucial to identify potential avenues for advancing our understanding and management of the FBR. As previously mentioned, the FBR involves a complex interplay between immune cells and implanted biomaterials. Effective regulation of this interplay is paramount for the success of biomaterial applications. There are several promising directions for future research.

One interesting method consist of co-delivering non-steroidal anti-inflammatory drugs (NSAIDs) alongside implants [32]. The underlying theory suggests that inhibiting the initial stage or acute inflammation stage of the FBR could be beneficial in preventing the deterioration of biomaterials utilized in tissue engineering. Although this approach has demonstrated effectiveness, it is not a long-term solution, as fibrosis and biomaterial degradation still occur over time [32].

Another novel approach involves co-delivering corticosteroids or tyrosine kinase inhibitors in conjunction with biomaterial implants. This innovative idea has exhibited promise in the context of implantation, offering an anti-fibrotic effect that could prove advantageous in the realm of wound healing. These drugs may play a role in reducing levels of TGF-B, thereby diminishing the immune response [32].

Immunomodulatory strategies represent an area of focus for orchestrating the host immune response to implanted biomaterials. This strategy includes exploring bioactive coatings, surface modifications, and functionalization techniques to minimize adverse immune reactions [67]. While immunomodulatory strategies for mitigating the FBR have proven to be highly effective in controlled research settings, their successful implementation in the clinical setting is still being validated [67].

### Future Direction of Biomaterial Design

Biomaterial design and characterization is an area of advancement that holds significant potential for minimizing the FBR. Integration of manufacturing techniques, such as 3D-printing and nanotechnology, can facilitate the fabrication of biomaterials with precisely controlled physical, chemical, and mechanical properties to optimize the biomaterial and host immune interaction [10,68]. Advancements in biomaterial design have yielded promising experimental results in successfully promoting tissue integration, and their clinical efficacy continues to evolve and undergo improvements [68].

The utilization of zwitterionic biomaterials has proven advantageous in addressing the FBR. These biomaterials are characterized by their unique surface composition, featuring both positive and negative charges [32,69]. This distinctive property often results in resistance to unspecific protein adhesion and cellular attachment. A study conducted by Zhang and colleagues provided evidence that zwitterionic hydrogels can stimulate angiogenesis, thus promoting enhanced wound healing while concurrently reducing the FBR [69]. Their dual charge nature, with both positive and negative charges, renders these hydrogels resistant to biofilm formation, bacterial proliferation, and the formation of capsules, even when implanted for a duration of up to three months [69].

Moreover, the reduction of the FBR through improved biomaterial components may frequently be observed in small animals such as rodents, rabbits, and dogs. However, to address the FBR challenge in clinical settings, further investigations involving larger animals is necessary. Consequently, continued exploration with optimized design advancements in diverse clinical contexts will be imperative for accelerating the field of regenerative medicine (Figure 3).

## 6. Conclusions

The study of FBR is a dynamic field of scientific investigation focused on enhancing care for individuals requiring biomaterial implants for various purposes such as wound healing, surgical procedures, prosthetics, diagnostic devices, and cosmetic enhancements. Researchers are actively exploring immunomodulation techniques, including therapeutic drug treatments and biomaterial coatings, to reduce immune responses to implanted foreign substances. Additionally, tissue engineering methods are being developed to design implants that seamlessly merge with the body.

## Figures and Tables

**Figure 1 bioengineering-10-01411-f001:**
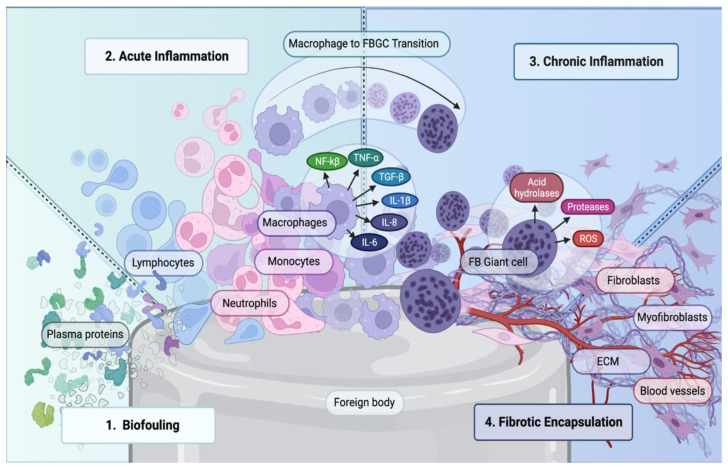
Immune-focused foreign body reaction. This figure illustrates the different steps of the foreign body response due to biomaterial implantation. Plasma proteins within the body interact with the biomaterial triggering the initiation of immune cell recruitment, including lymphocytes, neutrophils, monocytes and eventually macrophages. Macrophages in turn release a variety of cytokines such as NF-κB, TNF-α, TGF-β, IL-1β, IL-6, and IL-8, which further enhance the recruitment of macrophages, promote phagocytosis, and contribute FBGC. The presence of FBGC leads to the secretion of proteases, acid hydrolases, ROS, fibroblasts, and generation of ECM, ultimately resulting in the development of a fibrotic capsule surrounding biomaterial.

**Figure 2 bioengineering-10-01411-f002:**
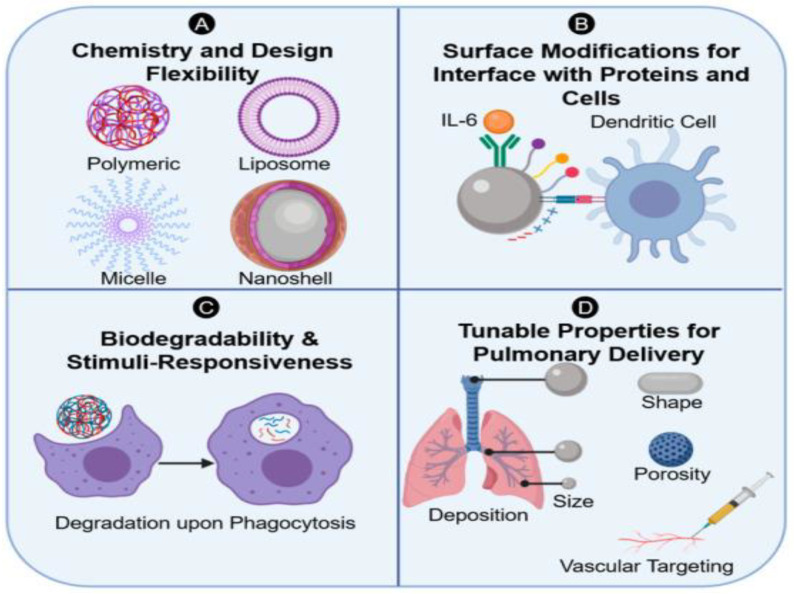
Tunable properties of biomaterials. This figure depicts different tunable properties of biomaterials that promote tissue integration. Modifications are made to the composition of biomaterials through various approaches. These include altering (**A**) chemistry composition and flexibility, (**B**) implementing surface modifications, (**C**) adjusting the biodegradability properties and (**D**) modifying the delivery methods of the biomaterial. These are some strategies that aim to optimize the biomaterial characteristics and interactions with the host tissue. Reproduced with permission of reference [27].

**Figure 3 bioengineering-10-01411-f003:**
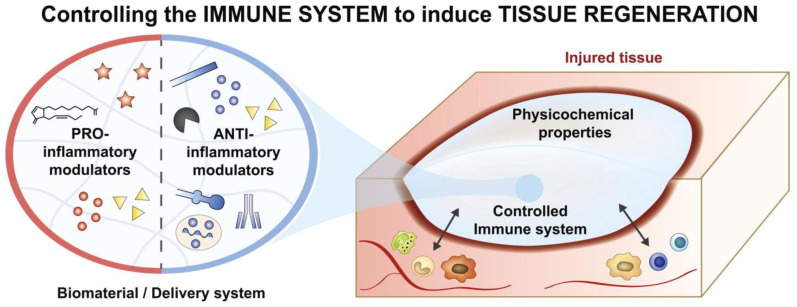
Immune system inducing tissue regeneration. This figure depicts the immunomodulatory properties of biomaterials to mitigate the foreign body response. The immune system releases proinflammatory cytokines and anti-inflammatory cytokines to increase tissue regeneration to increase wound healing [70].

**Table 1 bioengineering-10-01411-t001:** Key cytokines secreted by macrophages. This table illustrates the cytokines released by macrophages and their function within the foreign body response.

Cytokine	Function	Reference
Tumor necrosis factor-alpha (TNF-α)	Induces inflammation through activation of proinflammatory signaling pathways; activates apoptotic death	Ksontini et al. [16]
Interleukin-1β (IL-1β)	Induces inflammation through promoting recruitment and proliferation of innate immune cells; induces differentiation of type 17 T-helper (TH17) cells	Gabay et al. [17] Bent et al. [18]
Interleukin-6 (IL-6)	Induces inflammation through inducing synthesis of acute phase proteins and antibody production for elimination of infectious agents; supports differentiation of effector T cells	Tanaka et al. [19] Tanaka et al. [20]
Interleukin-8 (IL-8)	Inducing inflammation through recruiting and activating neutrophils	Harada et al. [21]
Interleukin-4, Interleukin-13 (IL-4, IL-13)	Prevents inflammation through directly inducing alternative (M2) activation of macrophages; promoting Th2 cell type response; and driving profibrotic activation and foreign body cell formation	Gordon & Martinez [22]
Interleukin-10 (IL-10)	Prevents inflammation through downregulating macrophage gene expression; directing Treg-mediated anti-inflammatory response; and preventing scar formation	Zhang et al. [14], Gordon & Martinez [22]

**Table 2 bioengineering-10-01411-t002:** Proteins involved in FBR.

Protein	Function	Reference
Cytokines/Chemokines	Recruit immune response by recruiting immune cells to the implant site and promoting inflammation.	Veiseh et al. [32]
Fibronectin	A glycoprotein that can bind to surface of biomaterials and facilitate the adhesion of immune cells, particularly macrophages. This protein is involved in the initial recognition of the foreign material	Veiseh et al. [32]
Collagen	Collagen is a major component of the extracellular matrix and is produced during the FBR. It contributes to the formation of a fibrous capsule around the foreign material, isolating it from its surroundings.	Veiseh et al. [32]
TGF-β	TGF-β is a cytokine that plays a key role in tissue repair and fibrosis. Its production is often increased during the FBR and can contribute to the development of fibrous tissue around the implant.	Veiseh et al. [32]
Plasma Proteins	Various proteins such as fibrinogen can absorb into the surface of biomaterials promoting protein adhesion and cell attachment.	Veiseh et al. [32]

**Table 3 bioengineering-10-01411-t003:** Growth factors in wound healing.

Growth Factors	Function	Reference
PDGF	PDGF is released by platelets and various cell types during the early stages of tissue injury and the FBR. It stimulates cell migration, proliferation, and the production of extracellular matrix components such as collagen, contributing to tissue repair fibrosis.	Yamakawa S. [55]
VEGF	VEGF is a critical growth factor in angiogenesis. It promotes the formation of new blood vessels, which can be important for supplying nutrients and oxygen to tissues surrounding the implant.	Yamakawa S. [55]
bFGFs	bFGFs are a family of growth factors that promote cell growth and angiogenesis. They can influence the tissue response of implanted materials by stimulation fibroblasts and endothelial.	Yamakawa S. [55]
TGF-β	TGF-β is involved in many aspects of wound healing. It promotes cell migration, angiogenesis, collagen synthesis, and tissue remodeling. There are multiple isoforms of TGF-β, and they can have distinct roles in different phases of wound healing.	Yamakawa S. [55]

## Data Availability

Not applicable.

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
