# Peer review of "Investigating Immunomodulatory Biomaterials for Preventing the Foreign Body Response"

_bioengineering, 2023, doi:10.3390/bioengineering10121411_

Round 1
Reviewer 1 Report (Previous Reviewer 2)
Comments and Suggestions for Authors
This is a re-submitted manuscript that I’ve reviewed previously.
There are some revisions made in this version. The manuscript is well-organized and structured. The focus is on the foreign body response (FBR) due to the introduction of medical implants. A much-simplified wording of the topic is the control of anti-inflammation within tissues. Since this is a min-review, I suggest several points to improve the clarity of overall writing:
1. Figure 1 illustrates the different stages of the FBR due to implants. The four stages from biofouling to acute inflammation, chronic inflammation, and final fibrotic encapsulation can be sequential. But is this sequence clearly defined or is it can move from one stage to another by skipping one, for example, from acute inflammation to fibrotic encapsulation directly?
2. The suppression of cytokines is important to control the inflammatory response. It is more instructive to readers if the authors can put up a short, more quantitative discussion about the changes among cytokines in Table 1 to boost or reduce the inflammatory response. This could help readers better understand if they would like to work on the issue of anti-inflammatory control.
3. Similarly, discussions on the quantitative changes among the key biomarkers representing the shift from M1 to M2 phases of macrophages or vice versa benefit many readers. (4.3. Protein-Functionalized Biomaterials).
Comments on the Quality of English LanguageNone
Author Response
Reviewer 1
There are some revisions made in this version. The manuscript is well-organized and structured. The focus is on the foreign body response (FBR) due to the introduction of medical implants. A much-simplified wording of the topic is the control of anti-inflammation within tissues. Since this is a min-review, I suggest several points to improve the clarity of overall writing:
The authors thank reviewer 1 for their careful read of our manuscript and are pleased to hear that this reviewer has found our paper to be well-organized and structured. We thank the reviewer for their additional comments below and have responded to each below.
- Figure 1 illustrates the different stages of the FBR due to implants. The four stages from biofouling to acute inflammation, chronic inflammation, and final fibrotic encapsulation can be sequential. But is this sequence clearly defined or is it can move from one stage to another by skipping one, for example, from acute inflammation to fibrotic encapsulation directly?
The authors thank the reviewer for this comment, which highlights the important process of FBR. The reviewer is correct that these four stages outlined in the figure are sequential and formation of FBR universally includes each of these stages in some capacity. Language has been added to clarify this important point. The paragraph on page 2 now reads as follows:
The FBR is a complex series of biological events that occur when a biomaterial or foreign object is implanted into the body. While the exact sequence and duration of these events may vary depending on factors such as the type of biomaterial and the individual response, the FBR represents an overall inflammatory process involving protein adsorption, neutrophil and macrophage activation, foreign body giant cell formation, and fibrotic encapsulation surrounding the implanted biomaterial (Figure 1) [10,11]. While these stages are dynamic and may occur concurrently, each has been recorded consistently in the formation of FBR [10, 11, 12]. When a biomaterial is surgically introduced into the body, an immunological cascade is activated. The biomaterial may be a medical device, implant, scaffold, or any foreign object designed for biomedical applications [10,12]. Upon contact with the surrounding tissue and blood, the biomaterial interacts with the host plasma components, leading to adsorption of proteins, such as fibrinogen, albumin, and immunoglobulins, onto its surface [12].
- The suppression of cytokines is important to control the inflammatory response. It is more instructive to readers if the authors can put up a short, more quantitative discussion about the changes among cytokines in Table 1 to boost or reduce the inflammatory response. This could help readers better understand if they would like to work on the issue of anti-inflammatory control.
The authors thank the reviewer for this inciteful recommendation to clarify the inflammatory or anti-inflammatory action of the relevant cytokines summarized in Table 1. We have added clarifying language to the table, shown below.
|
Cytokine |
Function |
Reference |
|
Tumor necrosis factor-alpha (TNF-α) |
Induces inflammation through activation of proinflammatory signaling pathways; activates apoptotic death |
Ksontini et al. [16] |
|
Interleukin-1β (IL-1β) |
Induces inflammation through promoting recruitment and proliferation of innate immune cells; induces differentiation of type 17 T-helper (TH17) cells |
Gabay et al. [17] Bent et al. [18] |
|
Interleukin-6 (IL-6) |
Induces inflammation through inducing synthesis of acute phase proteins and antibody production for elimination of infectious agents; supports differentiation of effector T cells |
Tanaka et al. [19] Tanaka et al. [20] |
|
Interleukin-8 (IL-8) |
Inducing inflammation through recruiting and activating neutrophils |
Harada et al. [21] |
|
Interleukin-4, Interleukin-13 (IL-4, IL-13) |
Prevents inflammation through directly inducing alternative (M2) activation of macrophages; promoting Th2 cell type response; driving profibrotic activation and foreign body cell formation |
Gordon & Martinez [22] |
|
Interleukin-10 (IL-10) |
Prevents inflammation through downregulating macrophage gene expression; directing Treg-mediated anti-inflammatory response; preventing scar formation |
Zhang et al. [14], Gordon & Martinez [22] |
- Similarly, discussions on the quantitative changes among the key biomarkers representing the shift from M1 to M2 phases of macrophages or vice versa benefit many readers. (4.3. Protein-Functionalized Biomaterials).
As above, the authors thank the reviewer for noting the importance of clarification on these directional changes among key biomarkers. We agree that this information will improve the manuscript and we have added text to the paper to this effect, clarifying the impact of the M2 phenotype. We feel that other biomarkers and proteins discussed in this section are well-specified. The section now reads as follows:
Integrin-targeting peptides directly interact with immune cells in the wound environment to drive macrophage polarity towards an anti-inflammatory phenotype and shape multiple upstream signaling pathways [51]. Cha et al. observed that inhibition of integrin α2β1 blocks the induction of anti-inflammatory M2 macrophages polarization. This suggest that integrin peptides play an integral role in promoting a microenvironment that favors M2 polarity, reduces fibrosis, and enhances biocompatibility [51].
Specifically, integrin-binding arginine-glycine-aspartic acid (RGD) peptides, which are derived from fibronectin, have recently been integrated in biomaterial engineering as common immunomodulatory components [54]. Wu et al., utilized phosphatidylserine-containing liposomes (PSLs) with RGD motifs to drive M2 polarization of macrophages and promote bone regeneration.54 Overall, RGD-PSLs had a concerted immunomodulatory effect through enhanced M2 macrophage marker genes, such as Arg-1, FIZZ-1, and YM-1, and suppression of pro-inflammatory cytokine (IL-1β, IL-6, TNF-α) expression both in vitro and in vivo using a calvarial defect rat model [54].
Reviewer 2 Report (Previous Reviewer 1)
Comments and Suggestions for Authors
Manuscript Number: bioengineering-2671297
The manuscript entitled: Investigating Immunomodulatory Biomaterials for Preventing the Foreign Body Response by Kim A et al. reviewed scientific literature to deep the issue of implantable biomaterials and the complex biomaterial-host cell interactions that occur within the tissue microenvironment.
The authors made a relevant effort to improve the overall quality of the paper since many modifications were made in the manuscript text, addressing further comments in the letter.
Comments on the Quality of English Language
Manuscript Number: bioengineering-2671297
The reviewer considers it necessary for the authors to implement the English of the text by rephrasing some sentences and correcting typos.
Author Response
The manuscript entitled: Investigating Immunomodulatory Biomaterials for Preventing the Foreign Body Response by Kim A et al. reviewed scientific literature to deep the issue of implantable biomaterials and the complex biomaterial-host cell interactions that occur within the tissue microenvironment.
The authors made a relevant effort to improve the overall quality of the paper since many modifications were made in the manuscript text, addressing further comments in the letter.
The authors thank reviewer 2 for their thoughtful comments. We are pleased that this reviewer has found our revisions to improve the quality of the paper and others have been satisfactorily addressed.
Reviewer 3 Report (New Reviewer)
Comments and Suggestions for Authors
The manuscript entitled "Investigating Immunomodulatory Biomaterials for Preventing the Foreign Body Response"authored by Kim et al. presented an overview of foriegn body response, biomaterial design and effect of surface chemistry in particular on the immunomodulation. Overall, the review was crisp, focused and well-presented.
Author Response
Reviewer 3
The manuscript entitled "Investigating Immunomodulatory Biomaterials for Preventing the Foreign Body Response" authored by Kim et al. presented an overview of foreign body response, biomaterial design and effect of surface chemistry in particular on the immunomodulation. Overall, the review was crisp, focused, and well-presented.
The authors thank reviewer 3 for their thoughtful comments. We are pleased to hear this reviewer found our manuscript to be crisp, focused, and well-presented.
This manuscript is a resubmission of an earlier submission. The following is a list of the peer review reports and author responses from that submission.
Round 1
Reviewer 1 Report
Comments and Suggestions for Authors
Manuscript ID: bioengineering-2529880
The manuscript entitled “Investigating Immunomodulatory Biomaterials for Preventing the Foreign Body Response” by Alexia Kim et al. reviewed the current literature on the response to implant rejection in determining proinflammatory macrophages and fibroblasts activation that contribute to a synergistically destructive process of uncontrolled inflammation and excessive fibrosis. In addition, the authors provide a comprehensive explanation of the techniques used to overcome the immune response and modify the foreign body response for improved outcomes.
The manuscript is quite well written, however it needs to be further improved to better fit it as a review article to be published on Bioengineering.
In particular, the following parts should be really improved: lines 37-40, lines 45-54, lines 87-96, lines 161-168 and 211-218.
Moreover, it is unclear in which field the topic is reported: Orthopaedic? Dental? Cancer? Other. It should be mentioned in lines 66-68. These publications, for example, can be introduced to better answer to this concern:
doi: 10.1016/j.biomaterials.2021.121114
doi: 10.1177/039463201002300126
doi: 10.1038/s41577-021-00540-z
doi: 10.3389/fendo.2020.00386
Additionally, other aspects should be introduce into the manuscript: for example, a literature analysis of the biomaterials that are currently used in surgery and determined the FBR (lines 153-157), as the biomaterial chemical composition, such as type and structures/modifications (lines 251-262).
In line 178, the authors introduced the “biofilm” but no explanation of the microbial role in FBR is further explored: It should be done. As should be deepen the release of other compound from the biomaterials that could also influence the immune response (i.e. vitamin E, antibiotics, etc.).
The “future directions” section should be rewritten with a more specific approach; in fact, all the article is very superficial in its treatment, so at least this part should be more specific and detailed.
Finally, the figures reported into the text are of poor quality.
Author Response
Date: September 29th, 2023
Re: Investigating Immunomodulatory Biomaterials for Preventing the Foreign Body Response
Manuscript Number: bioengineering-2529880
Bioengineering Editors and Reviewers:
We thank the editors and reviewers for their careful consideration of our manuscript. Their thoughtful and constructive comments have led to a significantly revised and improved manuscript. Each reviewer’s comments are addressed below, including references to the corresponding changes to the manuscript.
Reviewer comments are provided below in bold black font. Our responses are provided in blue font. New text is in red font.
Thank you for your continued consideration of this manuscript.
Sincerely,
Dr Michelle Griffin MBChb PhD
Reviewer 1
- In particular, the following parts should be really improved: lines 37-40, lines 45-54, lines 87-96, lines 161-168 and 211-218.
Thank you for your comment and for your specific line suggestions for improvement. The team has made substantial edits to this manuscript with particular focus on the lines indicated. Changes have been made that we feel reflect substantial improvement.
- Moreover, it is unclear in which field the topic is reported: Orthopaedic? Dental? Cancer? Other. It should be mentioned in lines 66-68. These publications, for example, can be introduced to better answer to this concern:
doi: 10.1016/j.biomaterials.2021.121114
doi: 10.1177/039463201002300126
doi: 10.1038/s41577-021-00540-z
doi: 10.3389/fendo.2020.00386
Thank you for your comment. The manuscript was changed to focus on different aspects of surgery. These additions can be seen below and on pages 4 and 8.
Page 4
Ibrahim et al. provided evidence that the FBR exhibits a degree of specificity depending on the type of implant employed, resulting in variable levels of capsule thickness and cellular reactions.13 In their research, they inserted biocompatible materials including silicone sheets polyvinyl alcohol (PVA), expanded PTFE (EPTFE), Polypropylene and cotton sheets ( control materials), in order to investigate whether the use of biocompatible materials could help reduce the occurrence of FBR.13 This experiment demonstrated the capacity to amplify or diminish FBR-associated cellular activity. Notably, their specific findings indicated an increase in both cellular activity and capsular thickness when using cotton sheets. Silicone sheets formed the thinnest capsules, and PVA showed an increased number of Giant cells. Histological analysis using Masson’s trichrome stained sections demonstrated not statistically significant change between the implants Consequently, they concluded that meticulous consideration of biomaterial selection is essential to mitigate the foreign body response in biomaterial implantation. This example bears relevance to the realm of plastic surgery procedures involving silicone implantation.13
Page 8
In the field of plastic and reconstructive surgery and skin regeneration a wide array of biomaterials finds application in reconstructive procedures.26 Peng et al. conducted a comprehensive review encompassing various biomaterials utilized in plastic and reconstructive surgery. Among these materials, natural biomaterials comprise components such as animal-derived proteins, decellularized tissues, collagen, and hyaluronic acid. While natural biomaterials tend to evoke a minimal FBR, their practical application may be limited in scope or accompanied by high costs.26
Conversely, synthetic biomaterials are also employed extensively in surgical settings for reconstructive purposes.26 These materials encompass, but are not restricted to, substances like silicone, expanded polytetrafluoroethylene, polymethyl methacrylate, polylactic acid, and polyglycolic acid. Synthetic materials offer a broad spectrum of applications, including prostheses, 3D printing, sutures, and drug delivery. However, they carry the potential for eliciting a foreign body response such as capsular formation, biomaterial degradation, infection, and granuloma formation.26
Presently, researchers are actively engaged in efforts to integrate various materials alongside biomaterial implants in order to mitigate the immune response they may trigger.13
- Additionally, other aspects should be introduce into the manuscript: for example, a literature analysis of the biomaterials that are currently used in surgery and determined the FBR (lines 153-157), as the biomaterial chemical composition, such as type and structures/modifications (lines 251-262).
Thank you for your comment. The manuscript has been improved to include the analysis of different biomaterials. The added text has been included below as well as on page 8.
In the field of plastic and reconstructive surgery and skin regeneration a wide array of biomaterials finds application in reconstructive procedures.26 Peng et al. conducted a comprehensive review encompassing various biomaterials utilized in plastic and reconstructive surgery. Among these materials, natural biomaterials comprise components such as animal-derived proteins, decellularized tissues, collagen, and hyaluronic acid. While natural biomaterials tend to evoke a minimal FBR, their practical application may be limited in scope or accompanied by high costs.26
Conversely, synthetic biomaterials are also employed extensively in surgical settings for reconstructive purposes.26 These materials encompass, but are not restricted to, substances like silicone, expanded polytetrafluoroethylene, polymethyl methacrylate, polylactic acid, and polyglycolic acid. Synthetic materials offer a broad spectrum of applications, including prostheses, 3D printing, sutures, and drug delivery. However, they carry the potential for eliciting a foreign body response such as capsular formation, biomaterial degradation, infection, and granuloma formation.26
Presently, researchers are actively engaged in efforts to integrate various materials alongside biomaterial implants in order to mitigate the immune response they may trigger.13
- In line 178, the authors introduced the “biofilm” but no explanation of the microbial role in FBR is further explored: It should be done. As should be deepen the release of other compound from the biomaterials that could also influence the immune response (i.e. vitamin E, antibiotics, etc.).
Thank you for your comment. We agree with the reviewer and the article was updated to include more information on biofilms and the microbial role on FBR. This change can be seen on page 10 of the altered manuscript as well as below.
Immune system and biomaterial design
Advancements in biomaterial design are able to overcome implant rejection and promote tissue integration through tunable properties that either render the biomaterial immunologically inert or enhance their immunomodulatory capabilities. Biomaterials can be synthesized in a wide range of compositions, sizes, shapes, and porosities with surface functionalization specifically catered to the appropriate disease context, allowing for optimization and customization of design.27,28 By carefully designing implantable materials and surfaces with biocompatible parameters, biomaterials can be implemented in a way that maximizes their therapeutic potential and promotes wound healing in diverse contexts.
Specific modifications can significantly reduce FBR and evade immune recognition. Generally, biomaterials with decreased protein adsorption are developed to prevent cell adhesion and activation.29 Protein adsorption represents the crosstalk at the biomaterial-tissue interface that initiates the immune response and FBR. While monocytes and macrophages tend to adsorb on implant surfaces, the protein layer chemistry can be modified to target a specific amount and identity of the substrates that are adsorbed.30 Surfaces with poly(ethylene glycol) (PEG), for instance, minimize interactions at the interface, resulting in a significantly thinner fibrotic capsule surrounding the implant.31
Researchers have explored various methods to reduce biofilm formation in the context of biomaterial implants. Biofilms are structured communities of microorganisms that can attach to implant and medical device surfaces, act as protective layers for these microorganisms on the implanted material. This protection can lead to prolonged inflammation, infections, and complications, thereby intensifying the FBR. Zhi et al. showcased a reduction in biofilm formation on silicone rubber surfaces when coated with polyhexanide (PHMB) and higher molecular weight PEG, highlighting the anti-biotic properties of these coatings.32
- The “future directions” section should be rewritten with a more specific approach; in fact, all the article is very superficial in its treatment, so at least this part should be more specific and detailed.
Thank you for your comment. The manuscript’s future directions section of the manuscript has been rewritten, as suggested, with a more specific approach to include additional detail. This update can be seen below and on pages 16-18.
5.0 Future directions
As researchers delve deeper into the realm of biomaterials and their applications in medical contexts, it remains crucial to identify potential avenues for advancing our understanding and management of the FBR. As previously mentioned, the FBR involves a complex interplay between immune cells and implanted biomaterials. Effective regulation of this interplay is paramount for the success of biomaterial applications. There are several promising directions for future research.
One interesting method consist of co-delivering non-steroidal anti-inflammatory drugs (NSAIDs) alongside implants.66 The underlying theory suggests that inhibiting the initial stage or acute inflammation stage of the FBR could be beneficial in preventing the deterioration of biomaterials utilized in tissue engineering. Although this approach has demonstrated effectiveness, it is not a long-term solution, as fibrosis and biomaterial degradation still occur over time.66
Another novel approach involves co delivering corticosteroids or tyrosine kinase inhibitors in conjunction with biomaterial implants. This innovative idea has exhibited promise in the context of implantation, offering an anti-fibrotic effect that could prove advantageous in the realm of wound healing. These drugs may play a role in reducing levels of TGF-B, thereby diminishing the immune response.66
Immunomodulatory strategies represent an area of focus for orchestrating the host immune response to implanted biomaterials. This strategy includes exploring bioactive coatings, surface modifications, and functionalization techniques to minimize adverse immune reactions.67 While immunomodulatory strategies for mitigating the FBR have proven to be highly effective in controlled research settings, their successful implementation in the clinical setting is still being validated.67
5.1 Future Direction of Biomaterial design
Biomaterial design and characterization is an area of advancement that holds significant potential for minimizing the FBR. Integration of manufacturing techniques, such as 3D-printing and nanotechnology, can facilitate the fabrication of biomaterials with precisely controlled physical, chemical, and mechanical properties to optimize the biomaterial and host immune interaction.10,68 Advancements in biomaterial design have yielded promising experimental results in successfully promoting tissue integration, and their clinical efficacy continues to evolve and undergo improvements.68
The Utilization of zwitterionic biomaterials has proven advantageous in addressing the FBR. These biomaterials are characterized by their unique surface composition, featuring both positive and negative charges.66,69 This distinctive property often results in resistance to unspecific protein adhesion and cellular attachment. A study conducted by Zhang and colleagues provided evidence that zwitterionic hydrogels can stimulate angiogenesis, thus promoting enhanced wound healing while concurrently reducing the FBR.69 Their dual charge nature, with both positive and negative charges, renders these hydrogels resistant to biofilm formation, bacterial proliferation, and the formation of capsules, even when implanted for a duration of up to three months.69
Moreover, the reduction of the FBR through improved biomaterial components may frequently be observed in small animals such as rodents, rabbits, and dogs. However, to address the FBR challenge in clinical settings, further investigations involving larger animals is necessary. Consequently, continued exploration with optimized design advancements in diverse clinical contexts will be imperative for accelerating the field of regenerative medicine.
Reviewer 2 Report
Comments and Suggestions for Authors
Investigating Immunomodulatory Biomaterials for Preventing the Foreign Body Response
It is suggested that three tables may be helpful for readers to understand the macrophage-related immune responses in the wound healing
1. Main pro-inflammatory and anti-inflammatory cytokines
2. Major classes of proteins for protein-functionalization
3. Commonly used growth factors in biomaterials for wound healing.
4. Please check Lines 273-274:
Cha et al. observed that hydrogels containing cell adhesive sequence peptides? exhibited monocyte expression of adhesion molecules….
This succinct mini-review indicates the macrophages and phagocytosis-related immune interactions between implanted materials and immune responses in the wound healing process. The overall quality of the presentation is well organized and delivered.
Author Response
Date: September 29th, 2023
Re: Investigating Immunomodulatory Biomaterials for Preventing the Foreign Body Response
Manuscript Number: bioengineering-2529880
Bioengineering Editors and Reviewers:
We thank the editors and reviewers for their careful consideration of our manuscript. Their thoughtful and constructive comments have led to a significantly revised and improved manuscript. Each reviewer’s comments are addressed below, including references to the corresponding changes to the manuscript.
Reviewer comments are provided below in bold black font. Our responses are provided in blue font. New text is in red font.
Thank you for your continued consideration of this manuscript.
Sincerely,
Dr Michelle Griffin MBChb PhD
Reviewer 2
It is suggested that three tables may be helpful for readers to understand the macrophage-related immune responses in the wound healing
- Main pro-inflammatory and anti-inflammatory cytokines
Thank you for your comment, we agree that these are important molecules to touch on in our review, and have covered this topic in table 1, titled:
Table 1: Key Main pro-inflammatory and anti-inflammatory cytokines released by macrophages
|
Cytokine |
Function |
Reference |
|
Tumor necrosis factor-alpha (TNF-α) |
Activates proinflammatory signaling pathways; activates apoptotic death |
Ksontini et al.16 |
|
Interleukin-1β (IL-1β) |
Promotes recruitment and proliferation of innate immune cells; induces differentiation of type 17 T-helper (TH17) cells |
Gabay et al.17 Bent et al.18 |
|
Interleukin-6 (IL-6) |
Induces synthesis of acute phase proteins and antibody production for elimination of infectious agents; supports differentiation of effector T cells |
Tanaka et al.,19 Tanaka et al.20 |
|
Interleukin-8 (IL-8) |
Recruits and activates neutrophils |
Harada et al. 21 |
|
Interleukin-4, Interleukin-13 (IL-4, IL-13) |
Directly induce alternative (M2) activation of macrophages; promotes Th2 cell type response; drives profibrotic activation and foreign body cell formation |
Gordon & Martinez22 |
|
Interleukin-10 (IL-10) |
Downregulates macrophage gene expression; directs Treg-mediated anti-inflammatory response; prevents scar formation |
Zhang et al.14, Gordon & Martinez22 |
- Major classes of proteins for protein-functionalization
Thank you for your comment. Table 2 was created to describe proteins involved in the FBR.
Table 2. Proteins involved in FBR
|
Protein |
Function |
Reference |
|
Cytokines/ Chemokines |
Recruit immune response by recruiting immune cells to the implant site and promoting inflammation. |
66Veiseh et al. |
|
Fibronectin |
A glycoprotein that can bind to surface of biomaterials and facilitate the adhesion of immune cells particularly macrophages. This protein is involved in the initial recognition of the foreign material |
66Veiseh et al. |
|
Collagen |
Collagen is a major component of the extracellular matrix and is produced during the FBR. It contributes to the formation of a fibrous capsule around the foreign material, isolating it from surrounding. |
66Veiseh et al. |
|
TGF-β |
TGF-β is a cytokine that plays a key role in tissue repair and fibrosis. Its production is often increased during the FBR and can contribute to the development of fibrous tissue around the implant. |
66Veiseh et al. |
|
Plasma Proteins |
Various proteins such as fibrinogen, can absorb into surface of biomaterials promoting protein adhesion and cell attachment. |
66Veiseh et al. |
- Commonly used growth factors in biomaterials for wound healing
Thank you for your comment. Table 3 was created to describe GF involved in the FBR and tissue regeneration.
Table 3 Growth factors in Wound healing
|
Growth Factors |
Function |
Reference |
|
PDGF |
PDGF is released by platelets and various cell types during the early stages of tissue injury and the FBR. It stimulates cell migration, proliferation, and the production of extracellular matrix components such as collagen contributing to tissue repair fibrosis. |
55 Yamakawa S |
|
VEGF |
VEGF is critical growth factor in angiogenesis. It promotes the formation of new blood vessels which can be important for supplying nutrients and oxygen of tissues surrounding the implant. |
55Yamakawa S |
|
bFGFs |
bFGFs are a family of growth factors that promote cell growth and angiogenesis. They can influence the tissue response of implanted materials by stimulation fibroblasts and endothelial. |
55Yamakawa S |
|
TGF-β |
TGF-β is involved in many aspects of wound healing. It promotes cell migration, angiogenesis, collagen synthesis, and tissue remodeling. There are multiple isoforms of TGF-β, and they can have distinct roles I different phases of wound healing. |
55Yamakawa S |
- Please check Lines 273-274: Cha et al. observed that hydrogels containing cell adhesive sequence peptides?exhibited monocyte expression of adhesion molecules.
Thank you for your comment. We agree with the reviewer and have made the necessary changes as can be seen on Page 14 and below.
Cha et al. observed that inhibtion of integrin α2β1 blocks the induction of M2 macrophages polarization. This suggest that integrin peptides play an integral role in promoting a microenvironment that favors M2 polarity, reduces fibrosis, and enhances biocompatibility51.
Reviewer 3 Report
Comments and Suggestions for Authors
TITLE: Investigating Immunomodulatory Biomaterials for Preventing the Foreign Body Response
bioengineering-2529880
The aim of the present investigation was to assess the to provide a comprehensive explanation of the techniques used to overcome the immune response and modify the FBR for improved outcomes. GENERAL COMMENTS
The article is in-line with the journal topic, but severe flaws are present. The investigation topic is interesting, but the present paper need further interventions and improvement to increase the scientific soundness. The authors is not recommended for publication considering the high standard of the present journal and encouraged to resubmission after a consistent revision.
Title: The title should indicate the type of study that has been conducted. In this case the article could be considered as a short communication.
Introduction
1. The articles did not consider the preclinical and histological evidence regarding the foreign body response to the biomaterials. A separated paragraph (and a systematic methodology application!) is strongly recommended.
2. The authors should describe separately the different medical fields and regenerative contexts of biomaterial applications and the related topic. (f.e: maxillofacial surgery, orthopedics, skin regeneration, dermatology)
3. Moreover, the different tissue regeneration failure models should be described.
4. In my opinion, the authors could apply a systematic review method considering separately the findings in preclinical/ animal model and the evidence on human tissues, a risk of bias assessment, and a meta-regression statistical model if applicable.
5. In this case, the authors should consider the PRISMA guidelines.
Author Response
Date: September 29th, 2023
Re: Investigating Immunomodulatory Biomaterials for Preventing the Foreign Body Response
Manuscript Number: bioengineering-2529880
Bioengineering Editors and Reviewers:
We thank the editors and reviewers for their careful consideration of our manuscript. Their thoughtful and constructive comments have led to a significantly revised and improved manuscript. Each reviewer’s comments are addressed below, including references to the corresponding changes to the manuscript.
Reviewer comments are provided below in bold black font. Our responses are provided in blue font. New text is in red font.
Thank you for your continued consideration of this manuscript.
Sincerely,
Dr Michelle Griffin MBChb PhD
Reviewer 3
- The articles did not consider the preclinical and histological evidence regarding the foreign body response to the biomaterials. A separated paragraph (and a systematic methodology application!) is strongly recommended.
Thank you for your comment, we agree that the preclinical and histological evidence regarding FBR needs to be considered in our article. We have added additional text addressing this comment, as can be seen below and on page 4.
Ibrahim et al. provided evidence that the FBR exhibits a degree of specificity depending on the type of implant employed, resulting in variable levels of capsule thickness and cellular reactions.13 In their research, they inserted biocompatible materials including silicone sheets polyvinyl alcohol (PVA), expanded PTFE (EPTFE), Polypropylene and cotton sheets ( control materials), in order to investigate whether the use of biocompatible materials could help reduce the occurrence of FBR.13 This experiment demonstrated the capacity to amplify or diminish FBR-associated cellular activity. Notably, their specific findings indicated an increase in both cellular activity and capsular thickness when using cotton sheets. Silicone sheets formed the thinnest capsules, and PVA showed an increased number of Giant cells. Histological analysis using Masson’s trichrome stained sections demonstrated not statistically significant change between the implants while a FBR was present. Consequently, they concluded that meticulous consideration of biomaterial selection is essential to mitigate the foreign body response in biomaterial implantation. This example bears relevance to the realm of plastic surgery procedures involving silicone implantation.13
- The authors should describe separately the different medical fields and regenerative contexts of biomaterial applications and the related topic. (f.e: maxillofacial surgery, orthopedics, skin regeneration, dermatology)
Thank you for your comment, we agree that addressing different medical fields and regenerative contexts of biomaterial applications and immune response is important, and have added an additional sections which can be seen below and on pages 4 and 8.
Page 4
Ibrahim et al. provided evidence that the FBR exhibits a degree of specificity depending on the type of implant employed, resulting in variable levels of capsule thickness and cellular reactions.13 In their research, they inserted biocompatible materials including silicone sheets polyvinyl alcohol (PVA), expanded PTFE (EPTFE), Polypropylene and cotton sheets ( control materials), in order to investigate whether the use of biocompatible materials could help reduce the occurrence of FBR.13 This experiment demonstrated the capacity to amplify or diminish FBR-associated cellular activity. Notably, their specific findings indicated an increase in both cellular activity and capsular thickness when using cotton sheets. Silicone sheets formed the thinnest capsules, and PVA showed an increased number of Giant cells. Histological analysis using Masson’s trichrome stained sections demonstrated not statistically significant change between the implants Consequently, they concluded that meticulous consideration of biomaterial selection is essential to mitigate the foreign body response in biomaterial implantation. This example bears relevance to the realm of plastic surgery procedures involving silicone implantation.13
Page 8
In the field of plastic and reconstructive surgery and skin regeneration a wide array of biomaterials finds application in reconstructive procedures.26 Peng et al. conducted a comprehensive review encompassing various biomaterials utilized in plastic and reconstructive surgery. Among these materials, natural biomaterials comprise components such as animal-derived proteins, decellularized tissues, collagen, and hyaluronic acid. While natural biomaterials tend to evoke a minimal FBR, their practical application may be limited in scope or accompanied by high costs.26
Conversely, synthetic biomaterials are also employed extensively in surgical settings for reconstructive purposes.26 These materials encompass, but are not restricted to, substances like silicone, expanded polytetrafluoroethylene, polymethyl methacrylate, polylactic acid, and polyglycolic acid. Synthetic materials offer a broad spectrum of applications, including prostheses, 3D printing, sutures, and drug delivery. However, they carry the potential for eliciting a foreign body response such as capsular formation, biomaterial degradation, infection, and granuloma formation.26
Presently, researchers are actively engaged in efforts to integrate various materials alongside biomaterial implants in order to mitigate the immune response they may trigger.13
- Moreover, the different tissue regeneration failure models should be described.
Thank you for your comment. We have added additional text describing different tissue regeneration failure models, as requested. The relevant text can be seen below and on page 2.
1.0 Introduction
Biomaterials have extensive applications in the field of medicine, serving as vital components in various medical interventions and treatments. At the forefront of regenerative medicine, these applications include implants, drug delivery systems, wound dressings, diagnostic tools, surgical devices, and bioabsorbable materials.1,2 However, it is important to acknowledge that the use of biomaterials can result in implant rejection and trigger an inflammatory and profibrotic reaction known as the foreign body response (FBR), which compromises the functionality and long-term performance of the biomaterial used and can lead to failure of tissue regeneration.1
Tissue regeneration can be compromised due to various factors, including inflammation, extracellular matrix dysfunction, angiogenesis deficiency, mechanical stress, and fibrosis.3 Inflammation is a pivotal player in tissue repair and regeneration; however, an exaggerated or prolonged inflammatory response can result in scarring and hinder the regenerative process.3 The extracellular matrix (ECM) provides critical structural support to tissues and is indispensable for successful regeneration. A dysfunctional ECM can disrupt cell migration, proliferation, and differentiation, ultimately impeding wound healing. Adequate angiogenesis, the formation of blood vessels, is vital for supplying tissues with the necessary nutrients and oxygen during regeneration. Insufficient angiogenesis can hamper tissue health and lead to reduced healing potential.3 Fibrosis, characterized by the excessive formation of fibrous tissue in response to injury, can obstruct normal tissue regeneration by forming excessive scar tissue.3 FBR contains various aspects of tissue regeneration failure, encompassing the factors mentioned above1.
Therefore, to combat the FBR a better understanding of the fundamental mechanisms of fibrotic capsular formation around and implant is crucial for developing effective treatments. Macrophages play an important role in material recognition and are often found adhering to the surrounding surfaces of the foreign object.4
4 In my opinion, the authors could apply a systematic review method considering separately the findings in preclinical/ animal model and the evidence on human tissues, a risk of bias assessment, and a meta-regression statistical model if applicable. In this case, the authors should consider the PRISMA guidelines.
Thank you for your comment, the authors acknowledge the strength and potential benefits of applying a systematic review lens to this important topic, including reproducibility and topic specificity. However, our goal is to provide a broad overview of a heterogenous topic that resists the typical systematic review PICO framework. Rather than answer a narrowly defined questions, we aim to synthesize a broad topic which touches on many different questions and parameters relevant to biomaterials and immune responses. We agree with this reviewer that systematic reviews may be a useful next step in this field, though we do not feel that this article type supports our current goals.